# DESIGNING NEURAL NETWORK ARCHITECTURES USING REINFORCEMENT LEARNING

**Bowen Baker, Otkrist Gupta, Nikhil Naik & Ramesh Raskar**
Media Laboratory
Massachusetts Institute of Technology
Cambridge MA 02139, USA
`{bowen, otkrist, naik, raskar}@mit.edu`

## ABSTRACT

At present, designing convolutional neural network (CNN) architectures requires both human expertise and labor. New architectures are handcrafted by careful experimentation or modified from a handful of existing networks. We introduce MetaQNN, a meta-modeling algorithm based on reinforcement learning to automatically generate high-performing CNN architectures for a given learning task. The learning agent is trained to sequentially choose CNN layers using $Q$-learning with an $\epsilon$-greedy exploration strategy and experience replay. The agent explores a large but finite space of possible architectures and iteratively discovers designs with improved performance on the learning task. On image classification benchmarks, the agent-designed networks (consisting of only standard convolution, pooling, and fully-connected layers) beat existing networks designed with the same layer types and are competitive against the state-of-the-art methods that use more complex layer types. We also outperform existing meta-modeling approaches for network design on image classification tasks.

## 1 INTRODUCTION

Deep convolutional neural networks (CNNs) have seen great success in the past few years on a variety of machine learning problems (LeCun et al., 2015). A typical CNN architecture consists of several convolution, pooling, and fully connected layers. While constructing a CNN, a network designer has to make numerous design choices: the number of layers of each type, the ordering of layers, and the hyperparameters for each type of layer, e.g., the receptive field size, stride, and number of receptive fields for a convolution layer. The number of possible choices makes the design space of CNN architectures extremely large and hence, infeasible for an exhaustive manual search. While there has been some work (Pinto et al., 2009; Bergstra et al., 2013; Domhan et al., 2015) on automated or computer-aided neural network design, new CNN architectures or network design elements are still primarily developed by researchers using new theoretical insights or intuition gained from experimentation.

In this paper, we seek to automate the process of CNN architecture selection through a meta-modeling procedure based on reinforcement learning. We construct a novel $Q$-learning agent whose goal is to discover CNN architectures that perform well on a given machine learning task with no human intervention. The learning agent is given the task of sequentially picking layers of a CNN model. By discretizing and limiting the layer parameters to choose from, the agent is left with a finite but large space of model architectures to search from. The agent learns through random exploration and slowly begins to exploit its findings to select higher performing models using the $\epsilon$-greedy strategy (Mnih et al., 2015). The agent receives the validation accuracy on the given machine learning task as the reward for selecting an architecture. We expedite the learning process through repeated memory sampling using experience replay (Lin, 1993). We refer to this $Q$-learning based meta-modeling method as MetaQNN, which is summarized in Figure 1.[1]

We conduct experiments with a space of model architectures consisting of only standard convolution, pooling, and fully connected layers using three standard image classification datasets: CIFAR-10,

---

[1]For more information, model files, and code, please visit *https://bowenbaker.github.io/metaqnn/*

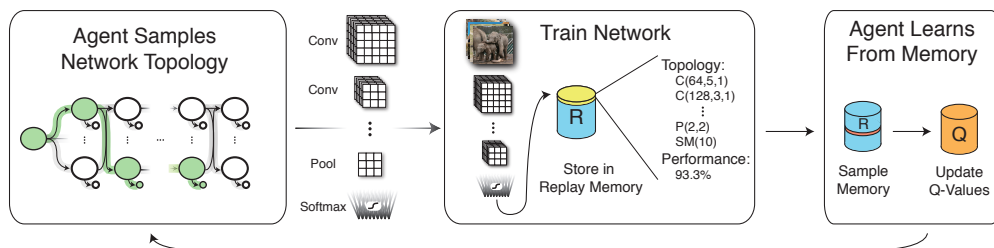

Figure 1: **Designing CNN Architectures with $Q$-learning:** The agent begins by sampling a Convolutional Neural Network (CNN) topology conditioned on a predefined behavior distribution and the agent's prior experience (left block). That CNN topology is then trained on a specific task; the topology description and performance, e.g. validation accuracy, are then stored in the agent's memory (middle block). Finally, the agent uses its memories to learn about the space of CNN topologies through $Q$-learning (right block).

SVHN, and MNIST. The learning agent discovers CNN architectures that beat all existing networks designed only with the same layer types (e.g., Springenberg et al. (2014); Srivastava et al. (2015)). In addition, their performance is competitive against network designs that include complex layer types and training procedures (e.g., Clevert et al. (2015); Lee et al. (2016)). Finally, the MetaQNN selected models comfortably outperform previous automated network design methods (Stanley & Miikkulainen, 2002; Bergstra et al., 2013). The top network designs discovered by the agent on one dataset are also competitive when trained on other datasets, indicating that they are suited for transfer learning tasks. Moreover, we can generate not just one, but several varied, well-performing network designs, which can be ensembled to further boost the prediction performance.

## 2 RELATED WORK

**Designing neural network architectures:** Research on automating neural network design goes back to the 1980s when genetic algorithm-based approaches were proposed to find both architectures and weights (Schaffer et al., 1992). However, to the best of our knowledge, networks designed with genetic algorithms, such as those generated with the NEAT algorithm (Stanley & Miikkulainen, 2002), have been unable to match the performance of hand-crafted networks on standard benchmarks (Verbancsics & Harguess, 2013). Other biologically inspired ideas have also been explored; motivated by screening methods in genetics, Pinto et al. (2009) proposed a high-throughput network selection approach where they randomly sample thousands of architectures and choose promising ones for further training. In recent work, Saxena & Verbeek (2016) propose to sidestep the architecture selection process through densely connected networks of layers, which come closer to the performance of hand-crafted networks.

Bayesian optimization has also been used (Shahriari et al., 2016) for automatic selection of network architectures (Bergstra et al., 2013; Domhan et al., 2015) and hyperparameters (Snoek et al., 2012; Swersky et al., 2013). Notably, Bergstra et al. (2013) proposed a meta-modeling approach based on Tree of Parzen Estimators (TPE) (Bergstra et al., 2011) to choose both the type of layers and hyperparameters of feed-forward networks; however, they fail to match the performance of hand-crafted networks.

**Reinforcement Learning:** Recently there has been much work at the intersection of reinforcement learning and deep learning. For instance, methods using CNNs to approximate the $Q$-learning utility function (Watkins, 1989) have been successful in game-playing agents (Mnih et al., 2015; Silver et al., 2016) and robotic control (Lillicrap et al., 2015; Levine et al., 2016). These methods rely on phases of *exploration*, where the agent tries to learn about its environment through sampling, and *exploitation*, where the agent uses what it learned about the environment to find better paths. In traditional reinforcement learning settings, over-exploration can lead to slow convergence times, yet over-exploitation can lead to convergence to local minima (Kaelbling et al., 1996). However, in the case of large or continuous state spaces, the $\epsilon$-*greedy strategy* of learning has been empirically shown to converge (Vermorel & Mohri, 2005). Finally, when the state space is large or exploration is costly,

the experience replay technique (Lin, 1993) has proved useful in experimental settings (Adam et al., 2012; Mnih et al., 2015). We incorporate these techniques—$Q$-learning, the $\epsilon$-greedy strategy and experience replay—in our algorithm design.

## 3 BACKGROUND

Our method relies on $Q$-learning, a type of reinforcement learning. We now summarize the theoretical formulation of $Q$-learning, as adopted to our problem. Consider the task of teaching an agent to find optimal paths as a Markov Decision Process (MDP) in a finite-horizon environment. Constraining the environment to be finite-horizon ensures that the agent will deterministically terminate in a finite number of time steps. In addition, we restrict the environment to have a discrete and finite state space $\mathcal{S}$ as well as action space $\mathcal{U}$. For any state $s_i \in \mathcal{S}$, there is a finite set of actions, $\mathcal{U}(s_i) \subseteq \mathcal{U}$, that the agent can choose from. In an environment with stochastic transitions, an agent in state $s_i$ taking some action $u \in \mathcal{U}(s_i)$ will transition to state $s_j$ with probability $p_{s'|s,u}(s_j|s_i, u)$, which may be unknown to the agent. At each time step $t$, the agent is given a reward $r_t$, dependent on the transition from state $s$ to $s'$ and action $u$. $r_t$ may also be stochastic according to a distribution $p_{r|s',s,u}$. The agent's goal is to maximize the total expected reward over all possible trajectories, i.e., $\max_{\mathcal{T}_i \in \mathcal{T}} R_{\mathcal{T}_i}$, where the total expected reward for a trajectory $\mathcal{T}_i$ is

$$R_{\mathcal{T}_i} = \sum_{(s,u,s') \in \mathcal{T}_i} \mathbb{E}_{r|s,u,s'}[r|s, u, s']. \tag{1}$$

Though we limit the agent to a finite state and action space, there are still a combinatorially large number of trajectories, which motivates the use of *reinforcement learning*. We define the maximization problem recursively in terms of subproblems as follows. For any state $s_i \in \mathcal{S}$ and subsequent action $u \in \mathcal{U}(s_i)$, we define the maximum total expected reward to be $Q^*(s_i, u)$. $Q^*(\cdot)$ is known as the *action-value* function and individual $Q^*(s_i, u)$ are know as $Q$-*values*. The recursive maximization equation, which is known as Bellman's Equation, can be written as

$$Q^*(s_i, u) = \mathbb{E}_{s_j|s_i,u}\left[\mathbb{E}_{r|s_i,u,s_j}[r|s_i, u, s_j] + \gamma \max_{u' \in \mathcal{U}(s_j)} Q^*(s_j, u')\right]. \tag{2}$$

In many cases, it is impossible to analytically solve Bellman's Equation (Bertsekas, 2015), but it can be formulated as an iterative update

$$Q_{t+1}(s_i, u) = (1 - \alpha)Q_t(s_i, u) + \alpha\left[r_t + \gamma \max_{u' \in \mathcal{U}(s_j)} Q_t(s_j, u')\right]. \tag{3}$$

Equation 3 is the simplest form of $Q$-learning proposed by Watkins (1989). For well formulated problems, $\lim_{t \to \infty} Q_t(s, u) = Q^*(s, u)$, as long as each transition is sampled infinitely many times (Bertsekas, 2015). The update equation has two parameters: (i) $\alpha$ is a $Q$-*learning rate* which determines the weight given to new information over old information, and (ii) $\gamma$ is the *discount factor* which determines the weight given to short-term rewards over future rewards. The $Q$-learning algorithm is *model-free*, in that the learning agent can solve the task without ever explicitly constructing an estimate of environmental dynamics. In addition, $Q$-learning is *off policy*, meaning it can learn about optimal policies while exploring via a non-optimal behavioral distribution, i.e. the distribution by which the agent explores its environment.

We choose the behavior distribution using an $\epsilon$-greedy strategy (Mnih et al., 2015). With this strategy, a random action is taken with probability $\epsilon$ and the greedy action, $\max_{u \in \mathcal{U}(s_i)} Q_t(s_i, u)$, is chosen with probability $1 - \epsilon$. We anneal $\epsilon$ from $1 \to 0$ such that the agent begins in an *exploration* phase and slowly starts moving towards the *exploitation* phase. In addition, when the exploration cost is large (which is true for our problem setting), it is beneficial to use the *experience replay* technique for faster convergence (Lin, 1992). In experience replay, the learning agent is provided with a memory of its past explored paths and rewards. At a given interval, the agent samples from the memory and updates its $Q$-values via Equation 3.

## 4 DESIGNING NEURAL NETWORK ARCHITECTURES WITH $Q$-LEARNING

We consider the task of training a learning agent to sequentially choose neural network layers. Figure 2 shows feasible state and action spaces (a) and a potential trajectory the agent may take along with the CNN architecture defined by this trajectory (b). We model the layer selection process as a Markov Decision Process with the assumption that a well-performing layer in one network should

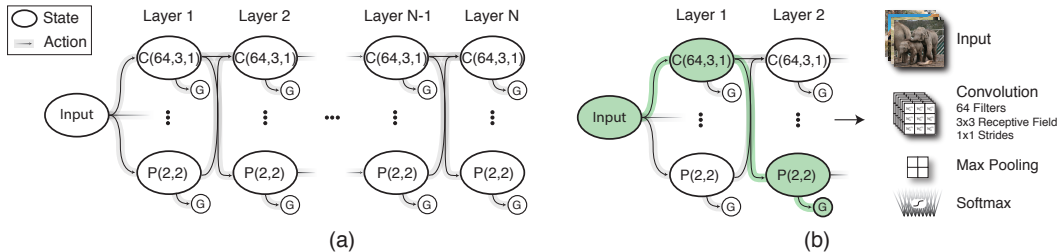

Figure 2: **Markov Decision Process for CNN Architecture Generation:** Figure 2(a) shows the full state and action space. In this illustration, actions are shown to be deterministic for clarity, but they are stochastic in experiments. $C(n, f, l)$ denotes a convolutional layer with $n$ filters, receptive field size $f$, and stride $l$. $P(f, l)$ denotes a pooling layer with receptive field size $f$ and stride $l$. $G$ denotes a termination state (Softmax/Global Average Pooling). Figure 2(b) shows a path the agent may choose, highlighted in green, and the corresponding CNN topology.

also perform well in another network. We make this assumption based on the hierarchical nature of the feature representations learned by neural networks with many hidden layers (LeCun et al., 2015). The agent sequentially selects layers via the $\epsilon$-greedy strategy until it reaches a termination state. The CNN architecture defined by the agent's path is trained on the chosen learning problem, and the agent is given a reward equal to the validation accuracy. The validation accuracy and architecture description are stored in a replay memory, and experiences are sampled periodically from the replay memory to update $Q$-values via Equation 3. The agent follows an $\epsilon$ schedule which determines its shift from exploration to exploitation.

Our method requires three main design choices: (i) reducing CNN layer definitions to simple state tuples, (ii) defining a set of actions the agent may take, i.e., the set of layers the agent may pick next given its current state, and (iii) balancing the size of the state-action space—and correspondingly, the model capacity—with the amount of exploration needed by the agent to converge. We now describe the design choices and the learning process in detail.

## 4.1 THE STATE SPACE

Each state is defined as a tuple of all relevant layer parameters. We allow five different types of layers: convolution (C), pooling (P), fully connected (FC), global average pooling (GAP), and softmax (SM), though the general method is not limited to this set. Table 1 shows the relevant parameters for each layer type and also the discretization we chose for each parameter. Each layer has a parameter *layer depth* (shown as Layer $1, 2, ...$ in Figure 2). Adding *layer depth* to the state space allows us to constrict the action space such that the state-action graph is directed and acyclic (DAG) and also allows us to specify a maximum number of layers the agent may select before terminating.

Each layer type also has a parameter called *representation size* ($R$-size). Convolutional nets progressively compress the representation of the original signal through pooling and convolution. The presence of these layers in our state space may lead the agent on a trajectory where the intermediate signal representation gets reduced to a size that is too small for further processing. For example, five $2 \times 2$ pooling layers each with stride 2 will reduce an image of initial size $32 \times 32$ to size $1 \times 1$. At this stage, further pooling, or convolution with receptive field size greater than 1, would be meaningless and degenerate. To avoid such scenarios, we add the $R$-*size* parameter to the state tuple $s$, which allows us to restrict actions from states with $R$-size $n$ to those that have a receptive field size less than or equal to $n$. To further constrict the state space, we chose to bin the representation sizes into three discrete buckets. However, binning adds uncertainty to the state transitions: depending on the true underlying representation size, a pooling layer may or may not change the $R$-size bin. As a result, the action of pooling can lead to two different states, which we model as stochasticity in state transitions. Please see Figure A1 in appendix for an illustrated example.

| Layer Type | Layer Parameters | Parameter Values |
|---|---|---|
| Convolution (C) | $i \sim$ Layer depth<br>$f \sim$ Receptive field size<br>$\ell \sim$ Stride<br>$d \sim$ # receptive fields<br>$n \sim$ Representation size | $< 12$<br>Square. $\in \{1, 3, 5\}$<br>Square. Always equal to 1<br>$\in \{64, 128, 256, 512\}$<br>$\in \{(\infty, 8], (8, 4], (4, 1]\}$ |
| Pooling (P) | $i \sim$ Layer depth<br>$(f, \ell) \sim$ (Receptive field size, Strides)<br>$n \sim$ Representation size | $< 12$<br>Square. $\in \big\{(5, 3), (3, 2), (2, 2)\big\}$<br>$\in \{(\infty, 8], (8, 4] \text{ and } (4, 1]\}$ |
| Fully Connected (FC) | $i \sim$ Layer depth<br>$n \sim$ # consecutive FC layers<br>$d \sim$ # neurons | $< 12$<br>$< 3$<br>$\in \{512, 256, 128\}$ |
| Termination State | $s \sim$ Previous State<br>$t \sim$ Type | <br>Global Avg. Pooling/Softmax |

Table 1: **Experimental State Space.** For each layer type, we list the relevant parameters and the values each parameter is allowed to take.

## 4.2 THE ACTION SPACE

We restrict the agent from taking certain actions to both limit the state-action space and make learning tractable. First, we allow the agent to terminate a path at any point, i.e. it may choose a termination state from any non-termination state. In addition, we only allow transitions for a state with layer depth $i$ to a state with layer depth $i + 1$, which ensures that there are no loops in the graph. This constraint ensures that the state-action graph is always a DAG. Any state at the maximum layer depth, as prescribed in Table 1, may only transition to a termination layer.

Next, we limit the number of fully connected (FC) layers to be at maximum two, because a large number of FC layers can lead to too may learnable parameters. The agent at a state with type FC may transition to another state with type FC if and only if the number of consecutive FC states is less than the maximum allowed. Furthermore, a state $s$ of type FC with number of neurons $d$ may only transition to either a termination state or a state $s'$ of type FC with number of neurons $d' \leq d$.

An agent at a state of type convolution (C) may transition to a state with any other layer type. An agent at a state with layer type pooling (P) may transition to a state with any other layer type other than another P state because consecutive pooling layers are equivalent to a single, larger pooling layer which could lie outside of our chosen state space. Furthermore, only states with representation size in bins $(8, 4]$ and $(4, 1]$ may transition to an FC layer, which ensures that the number of weights does not become unreasonably huge. Note that a majority of these constraints are in place to enable faster convergence on our limited hardware (see Section 5) and not a limitation of the method in itself.

## 4.3 $Q$-LEARNING TRAINING PROCEDURE

For the iterative $Q$-learning updates (Equation 3), we set the $Q$-learning rate ($\alpha$) to 0.01. In addition, we set the discount factor ($\gamma$) to 1 to not over-prioritize short-term rewards. We decrease $\epsilon$ from 1.0 to 0.1 in steps, where the step-size is defined by the number of *unique* models trained (Table 2). At $\epsilon = 1.0$, the agent samples CNN architecture with a random walk along a uniformly weighted Markov chain. Every topology sampled by the agent is trained using the procedure described in Section 5, and the prediction performance of this network topology on the validation set is recorded. We train a larger number of models at $\epsilon = 1.0$ as compared to other values of $\epsilon$ to ensure that the agent has adequate time to *explore* before it begins to *exploit*. We stop the agent at $\epsilon = 0.1$ (and not at $\epsilon = 0$) to obtain a stochastic final policy, which generates perturbations of the global minimum.[2] Ideally, we want to identify several well-performing model topologies, which can then be ensembled to improve prediction performance.

During the entire training process (starting at $\epsilon = 1.0$), we maintain a *replay dictionary* which stores (i) the network topology and (ii) prediction performance on a validation set, for all of the sampled

---

[2]$\epsilon = 0$ indicates a completely deterministic policy. Because we would like to generate several good models for ensembling and analysis, we stop at $\epsilon = 0.1$, which represents a stochastic final policy.

| $\epsilon$ | 1.0 | 0.9 | 0.8 | 0.7 | 0.6 | 0.5 | 0.4 | 0.3 | 0.2 | 0.1 |
|---|---|---|---|---|---|---|---|---|---|---|
| # Models Trained | 1500 | 100 | 100 | 100 | 150 | 150 | 150 | 150 | 150 | 150 |

Table 2: $\epsilon$ **Schedule.** The learning agent trains the specified number of unique models at each $\epsilon$.

models. If a model that has already been trained is re-sampled, it is not re-trained, but instead the previously found validation accuracy is presented to the agent. After each model is sampled and trained, the agent randomly samples 100 models from the replay dictionary and applies the $Q$-value update defined in Equation 3 for all transitions in each sampled sequence. The $Q$-value update is applied to the transitions in temporally reversed order, which has been shown to speed up $Q$-values convergence (Lin, 1993).

## 5 EXPERIMENT DETAILS

During the model exploration phase, we trained each network topology with a quick and aggressive training scheme. For each experiment, we created a validation set by randomly taking 5,000 samples from the training set such that the resulting class distributions were unchanged. For every network, a dropout layer was added after every two layers. The $i^{th}$ dropout layer, out of a total $n$ dropout layers, had a dropout probability of $\frac{i}{2n}$. Each model was trained for a total of 20 epochs with the Adam optimizer (Kingma & Ba, 2014) with $\beta_1 = 0.9$, $\beta_2 = 0.999$, $\varepsilon = 10^{-8}$. The batch size was set to 128, and the initial learning rate was set to 0.001. If the model failed to perform better than a random predictor after the first epoch, we reduced the learning rate by a factor of 0.4 and restarted training, for a maximum of 5 restarts. For models that started learning (i.e., performed better than a random predictor), we reduced the learning rate by a factor of 0.2 every 5 epochs. All weights were initialized with Xavier initialization (Glorot & Bengio, 2010). Our experiments using Caffe (Jia et al., 2014) took 8-10 days to complete for each dataset with a hardware setup consisting of 10 NVIDIA GPUs.

After the agent completed the $\epsilon$ schedule (Table 2), we selected the top ten models that were found over the course of exploration. These models were then finetuned using a much longer training schedule, and only the top five were used for ensembling. We now provide details of the datasets and the finetuning process.

The **Street View House Numbers (SVHN)** dataset has 10 classes with a total of 73,257 samples in the original training set, 26,032 samples in the test set, and 531,131 additional samples in the extended training set. During the exploration phase, we only trained with the original training set, using 5,000 random samples as validation. We finetuned the top ten models with the original plus extended training set, by creating preprocessed training and validation sets as described by Lee et al. (2016). Our final learning rate schedule after tuning on validation set was 0.025 for 5 epochs, 0.0125 for 5 epochs, 0.0001 for 20 epochs, and 0.00001 for 10 epochs.

**CIFAR-10**, the 10 class tiny image dataset, has 50,000 training samples and 10,000 testing samples. During the exploration phase, we took 5,000 random samples from the training set for validation. The maximum layer depth was increased to 18. After the experiment completed, we used the same validation set to tune hyperparameters, resulting in a final training scheme which we ran on the entire training set. In the final training scheme, we set a learning rate of 0.025 for 40 epochs, 0.0125 for 40 epochs, 0.0001 for 160 epochs, and 0.00001 for 60 epochs, with all other parameters unchanged. During this phase, we preprocess using global contrast normalization and use moderate data augmentation, which consists of random mirroring and random translation by up to 5 pixels.

**MNIST**, the 10 class handwritten digits dataset, has 60,000 training samples and 10,000 testing samples. We preprocessed each image with global mean subtraction. In the final training scheme, we trained each model for 40 epochs and decreased learning rate every 5 epochs by a factor of 0.2. For further tuning details please see Appendix C.

## 6 RESULTS

**Model Selection Analysis:** From $Q$-learning principles, we expect the learning agent to improve in its ability to pick network topologies as $\epsilon$ reduces and the agent enters the exploitation phase. In

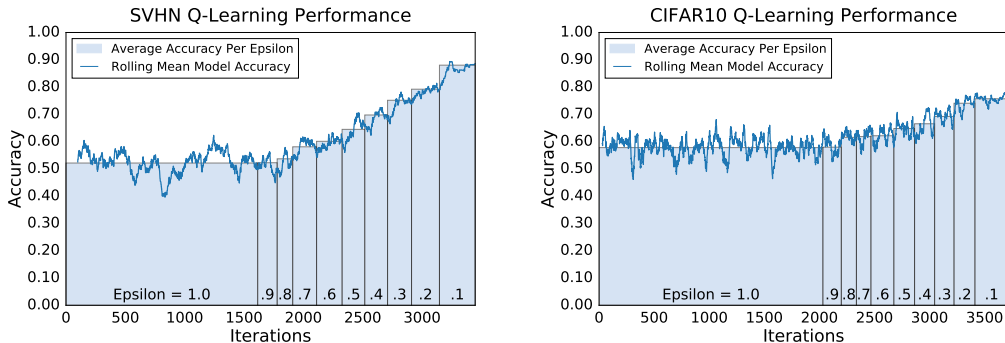

Figure 3: **Q-Learning Performance.** In the plots, the blue line shows a rolling mean of model accuracy versus iteration, where in each iteration of the algorithm the agent is sampling a model. Each bar (in light blue) marks the average accuracy over all models that were sampled during the exploration phase with the labeled $\epsilon$. As $\epsilon$ decreases, the average accuracy goes up, demonstrating that the agent learns to select better-performing CNN architectures.

| Method | CIFAR-10 | SVHN | MNIST | CIFAR-100 |
|---|---|---|---|---|
| Maxout (Goodfellow et al., 2013) | 9.38 | 2.47 | 0.45 | 38.57 |
| NIN (Lin et al., 2013) | 8.81 | 2.35 | 0.47 | 35.68 |
| FitNet (Romero et al., 2014) | 8.39 | 2.42 | 0.51 | 35.04 |
| HighWay (Srivastava et al., 2015) | 7.72 | - | - | - |
| VGGnet (Simonyan & Zisserman, 2014) | 7.25 | - | - | - |
| All-CNN (Springenberg et al., 2014) | 7.25 | - | - | 33.71 |
| MetaQNN (ensemble) | 7.32 | **2.06** | **0.32** | - |
| MetaQNN (top model) | **6.92** | 2.28 | 0.44 | **27.14*** |

Table 3: **Error Rate Comparison** with CNNs that only use convolution, pooling, and fully connected layers. We report results for CIFAR-10 and CIFAR-100 with moderate data augmentation and results for MNIST and SVHN without any data augmentation.

Figure 3, we plot the rolling mean of prediction accuracy over 100 models and the mean accuracy of models sampled at different $\epsilon$ values, for the CIFAR-10 and SVHN experiments. The plots show that, while the prediction accuracy remains flat during the exploration phase ($\epsilon = 1$) as expected, the agent consistently improves in its ability to pick better-performing models as $\epsilon$ reduces from 1 to 0.1. For example, the mean accuracy of models in the SVHN experiment increases from 52.25% at $\epsilon = 1$ to 88.02% at $\epsilon = 0.1$. Furthermore, we demonstrate the stability of the $Q$-learning procedure with 10 independent runs on a subset of the SVHN dataset in Section D.1 of the Appendix. Additional analysis of $Q$-learning results can be found in Section D.2.

The top models selected by the $Q$-learning agent vary in the number of parameters but all demonstrate high performance (see Appendix Tables 1-3). For example, the number of parameters for the top five CIFAR-10 models range from 11.26 million to 1.10 million, with only a 2.32% decrease in test error. We find design motifs common to the top hand-crafted network architectures as well. For example, the agent often chooses a layer of type $C(N, 1, 1)$ as the first layer in the network. These layers generate $N$ learnable linear transformations of the input data, which is similar in spirit to preprocessing of input data from RGB to a different color spaces such as YUV, as found in prior work (Sermanet et al., 2012; 2013).

**Prediction Performance:** We compare the prediction performance of the MetaQNN networks discovered by the $Q$-learning agent with state-of-the-art methods on three datasets. We report the accuracy of our best model, along with an ensemble of top five models. First, we compare MetaQNN with six existing architectures that are designed with standard convolution, pooling, and fully-connected layers alone, similar to our designs. As seen in Table 3, our top model alone, as well as the committee ensemble of five models, outperforms all similar models. Next, we compare our results with six top networks overall, which contain complex layer types and design ideas, including generalized pooling functions, residual connections, and recurrent modules. Our results are competitive with these methods as well (Table 4). Finally, our method outperforms existing automated network de-

| Method | CIFAR-10 | SVHN | MNIST | CIFAR-100 |
|---|---|---|---|---|
| DropConnect (Wan et al., 2013) | 9.32 | 1.94 | 0.57 | - |
| DSN (Lee et al., 2015) | 8.22 | 1.92 | 0.39 | 34.57 |
| R-CNN (Liang & Hu, 2015) | 7.72 | 1.77 | **0.31** | 31.75 |
| MetaQNN (ensemble) | 7.32 | 2.06 | 0.32 | - |
| MetaQNN (top model) | 6.92 | 2.28 | 0.44 | 27.14* |
| Resnet(110) (He et al., 2015) | 6.61 | - | - | - |
| Resnet(1001) (He et al., 2016) | **4.62** | - | - | **22.71** |
| ELU (Clevert et al., 2015) | 6.55 | - | - | 24.28 |
| Tree+Max-Avg (Lee et al., 2016) | 6.05 | **1.69** | **0.31** | 32.37 |

Table 4: **Error Rate Comparison** with state-of-the-art methods with complex layer types. We report results for CIFAR-10 and CIFAR-100 with moderate data augmentation and results for MNIST and SVHN without any data augmentation.

| Dataset | CIFAR-100 | SVHN | MNIST |
|---|---|---|---|
| Training from scratch | 27.14 | 2.48 | 0.80 |
| Finetuning | 34.93 | 4.00 | 0.81 |
| State-of-the-art | 24.28 (Clevert et al., 2015) | 1.69 (Lee et al., 2016) | 0.31 (Lee et al., 2016) |

Table 5: **Prediction Error** for the top MetaQNN (CIFAR-10) model trained for other tasks. Finetuning refers to initializing training with the weights found for the optimal CIFAR-10 model.

sign methods. MetaQNN obtains an error of 6.92% as compared to 21.2% reported by Bergstra et al. (2011) on CIFAR-10; and it obtains an error of 0.32% as compared to 7.9% reported by Verbancsics & Harguess (2013) on MNIST.

The difference in validation error between the top 10 models for MNIST was very small, so we also created an ensemble with all 10 models. This ensemble achieved a test error of **0.28%**—which beats the current state-of-the-art on MNIST without data augmentation.

The best CIFAR-10 model performs 1-2% better than the four next best models, which is why the ensemble accuracy is lower than the best model's accuracy. We posit that the CIFAR-10 MetaQNN did not have adequate exploration time given the larger state space compared to that of the SVHN experiment, causing it to not find more models with performance similar to the best model. Furthermore, the coarse training scheme could have been not as well suited for CIFAR-10 as it was for SVHN, causing some models to under perform.

**Transfer Learning Ability:** Network designs such as VGGnet (Simonyan & Zisserman, 2014) can be adopted to solve a variety of computer vision problems. To check if the MetaQNN networks provide similar transfer learning ability, we use the best MetaQNN model on the CIFAR-10 dataset for training other computer vision tasks. The model performs well (Table 5) both when training from random initializations, and finetuning from existing weights.

## 7 CONCLUDING REMARKS

Neural networks are being used in an increasingly wide variety of domains, which calls for scalable solutions to produce problem-specific model architectures. We take a step towards this goal and show that a meta-modeling approach using reinforcement learning is able to generate tailored CNN designs for different image classification tasks. Our MetaQNN networks outperform previous meta-modeling methods as well as hand-crafted networks which use the same types of layers.

While we report results for image classification problems, our method could be applied to different problem settings, including supervised (e.g., classification, regression) and unsupervised (e.g., autoencoders). The MetaQNN method could also aid constraint-based network design, by optimizing parameters such as size, speed, and accuracy. For instance, one could add a threshold in the state-action space barring the agent from creating models larger than the desired limit. In addition,

---

*Results in this column obtained with the top MetaQNN architecture for CIFAR-10, trained from random initialization with CIFAR-100 data.

one could modify the reward function to penalize large models for constraining memory or penalize slow forward passes to incentivize quick inference.

There are several future avenues for research in reinforcement learning-driven network design as well. In our current implementation, we use the same set of hyperparameters to train all network topologies during the $Q$-learning phase and further finetune the hyperparameters for top models selected by the MetaQNN agent. However, our approach could be combined with hyperparameter optimization methods to further automate the network design process. Moreover, we constrict the state-action space using coarse, discrete bins to accelerate convergence. It would be possible to move to larger state-action spaces using methods for $Q$-function approximation (Bertsekas, 2015; Mnih et al., 2015).

ACKNOWLEDGMENTS

We thank Peter Downs for creating the project website and contributing to illustrations. We acknowledge Center for Bits and Atoms at MIT for their help with computing resources. Finally, we thank members of Camera Culture group at MIT Media Lab for their help and support.

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

APPENDIX

# A ALGORITHM

We first describe the main components of the MetaQNN algorithm. Algorithm 1 shows the main loop, where the parameter $M$ would determine how many models to run for a given $\epsilon$ and the parameter $K$ would determine how many times to sample the replay database to update $Q$-values on each iteration. The function TRAIN refers to training the specified network and returns a validation accuracy. Algorithm 2 details the method for sampling a new network using the $\epsilon$-greedy strategy, where we assume we have a function TRANSITION that returns the next state given a state and action. Finally, Algorithm 3 implements the $Q$-value update detailed in Equation 3, with discounting factor set to 1, for an entire state sequence in temporally reversed order.

---

**Algorithm 1** $Q$-learning For CNN Topologies

---
**Initialize:**
 replay_memory $\leftarrow [\;]$
 $Q \leftarrow \{(s,u)\;\forall s \in \mathcal{S}, u \in \mathcal{U}(s)\;:\;0.5\}$
**for** episode = 1 to $M$ **do**
 $S, U \leftarrow$ SAMPLE_NEW_NETWORK($\epsilon$, $Q$)
 accuracy $\leftarrow$ TRAIN($S$)
 replay_memory.append((S, U, accuracy))
 **for** memory = 1 to $K$ **do**
 $S_{SAMPLE},\,U_{SAMPLE},\,$accuracy$_{SAMPLE} \leftarrow$ Uniform{replay_memory}
 $Q \leftarrow$ UPDATE_Q_VALUES($Q$, $S_{SAMPLE}$, $U_{SAMPLE}$, accuracy$_{SAMPLE}$)
 **end for**
**end for**

---

---

**Algorithm 2** SAMPLE_NEW_NETWORK($\epsilon$, $Q$)

---
**Initialize:**
 state sequence $S = [s_{\text{START}}]$
 action sequence $U = [\;]$
**while** $U[-1] \neq$ terminate **do**
 $\alpha \sim$ Uniform$[0, 1)$
 **if** $\alpha > \epsilon$ **then**
 $u = \text{argmax}_{u \in \mathcal{U}(S[-1])}\,Q[(S[-1], u)]$
 $s' = $ TRANSITION($S[-1], u$)
 **else**
 $u \sim$ Uniform$\{\mathcal{U}(S[-1])\}$
 $s' = $ TRANSITION($S[-1], u$)
 **end if**
 $U$.append($u$)
 **if** $u$ != terminate **then**
 $S$.append($s'$)
 **end if**
**end while**
**return** $S, U$

---

---

**Algorithm 3** UPDATE_Q_VALUES($Q$, $S$, $U$, accuracy)

---
$Q[S[-1], U[-1]] = (1 - \alpha)Q[S[-1], U[-1]] + \alpha \cdot$ accuracy
**for** i = length($S$) $- 2$ to 0 **do**
 $Q[S[i], U[i]] = (1 - \alpha)Q[S[i], U[i]] + \alpha \max_{u \in \mathcal{U}(S[i+1])} Q[S[i + 1], u]$
**end for**
**return** $Q$

---

## B    REPRESENTATION SIZE BINNING

As mentioned in Section 4.1 of the main text, we introduce a parameter called *representation size* to prohibit the agent from taking actions that can reduce the intermediate signal representation to a size that is too small for further processing. However, this process leads to uncertainties in state transitions, as illustrated in Figure A1, which is handled by the standard $Q$-learning formulation.

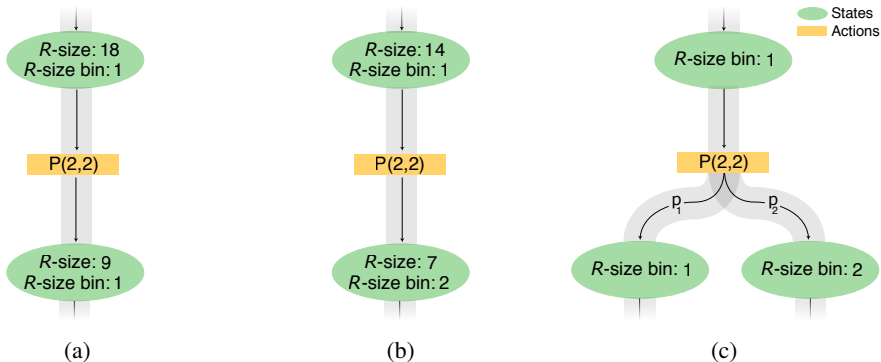

Figure A1: **Representation size binning:** In this figure, we show three example state transitions. The true *representation size* ($R$-size) parameter is included in the figure to show the true underlying state. Assuming there are two $R$-size bins, $R$-size $\text{Bin}_1$: $[8, \infty)$ and $R$-size $\text{Bin}_2$: $(0, 7]$, Figure A1a shows the case where the initial state is in $R$-size $\text{Bin}_1$ and true *representation size* is 18. After the agent chooses to pool with a $2 \times 2$ filter with stride 2, the true *representation size* reduces to 9 but the $R$-size bin does not change. In Figure A1b, the same $2 \times 2$ pooling layer with stride 2 reduces the actual *representation size* of 14 to 7, but the bin changes to $R$-size $\text{Bin}_2$. Therefore, in figures A1a and A1b, the agent ends up in different final states, despite originating in the same initial state and choosing the same action. Figure A1c shows that in our state-action space, when the agent takes an action that reduces the *representation size*, it will have uncertainty in which state it will transition to.

## C    MNIST EXPERIMENT

We noticed that the final MNIST models were prone to overfitting, so we increased dropout and did a small grid search for the weight regularization parameter. For both tuning and final training, we warmed the model with the learned weights from after the first epoch of initial training. The final models and solvers can be found on our project website *https://bowenbaker.github.io/metaqnn/*. Figure A2 shows the $Q$-Learning performance for the MNIST experiment.

## D    FURTHER ANALYSIS OF $Q$-LEARNING

Figure 3 of the main text and Figure A2 show that as the agent begins to exploit, it improves in architecture selection. It is also informative to look at the distribution of models chosen at each $\epsilon$. Figure A4 gives further insight into the performance achieved at each $\epsilon$ for both experiments.

### D.1    $Q$-LEARNING STABILITY

Because the $Q$-learning agent explores via a random or semi-random distribution, it is natural to ask whether the agent can consistently improve architecture performance. While the success of the three independent experiments described in the main text allude to stability, here we present further evidence. We conduct 10 independent runs of the $Q$-learning procedure on 10% of the SVHN dataset (which corresponds to $\sim$7,000 training examples). We use a smaller dataset to reduce the computation time of each independent run to 10GPU-days, as opposed to the 100GPU-days it would take on the full dataset. As can be seen in Figure A3, the $Q$-learning procedure with the exploration schedule detailed in Table 2 is fairly stable. The standard deviation at $\epsilon = 1$ is notably smaller than at other stages, which we attribute to the large difference in number of samples at each stage.

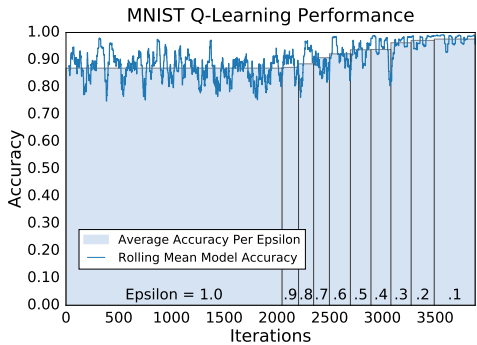

Figure A2: **MNIST $Q$-Learning Performance.** The blue line shows a rolling mean of model accuracy versus iteration, where in each iteration of the algorithm the agent is sampling a model. Each bar (in light blue) marks the average accuracy over all models that were sampled during the exploration phase with the labeled $\epsilon$. As $\epsilon$ decreases, the average accuracy goes up, demonstrating that the agent learns to select better-performing CNN architectures.

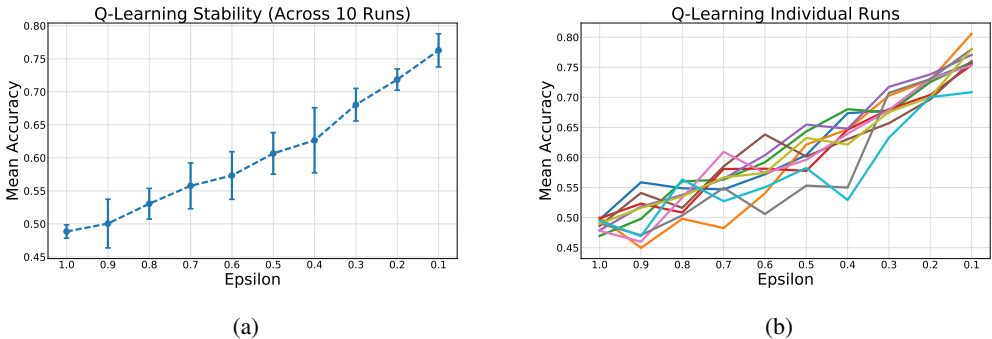

|     |     |
| --- | --- |
| (a) | (b) |

Figure A3: Figure A3a shows the mean model accuracy and standard deviation at each $\epsilon$ over 10 independent runs of the $Q$-learning procedure on 10% of the SVHN dataset. Figure A3b shows the mean model accuracy at each $\epsilon$ for each independent experiment. Despite some variance due to a randomized exploration strategy, each independent run successfully improves architecture performance.

Furthermore, the best model found during each run had remarkably similar performance with a mean accuracy of 88.25% and standard deviation of 0.58%, which shows that each run successfully found at least one very high performing model. Note that we did not use an extended training schedule to improve performance in this experiment.

## D.2  $Q$-VALUE ANALYSIS

We now analyze the actual $Q$-values generated by the agent during the training process. The learning agent iteratively updates the $Q$-values of each path during the $\epsilon$-greedy exploration. Each $Q$-value is initialized at 0.5. After the $\epsilon$-schedule is complete, we can analyze the final $Q$-value associated with each path to gain insights into the layer selection process. In the left column of Figure A5, we plot the average $Q$-value for each layer type at different layer depths (for both SVHN and CIFAR-10) datasets. Roughly speaking, a higher $Q$-value associated with a layer type indicates a higher probability that the agent will pick that layer type. In Figure A5, we observe that, while the average $Q$-value is higher for convolution and pooling layers at lower layer depths, the $Q$-values for fully-connected and termination layers (softmax and global average pooling) increase as we go deeper into the network. This observation matches with traditional network designs.

We can also plot the average $Q$-values associated with different layer parameters for further analysis. In the right column of Figure A5, we plot the average $Q$-values for convolution layers with receptive

field sizes 1, 3, and 5 at different layer depths. The plots show that layers with receptive field size of 5 have a higher $Q$-value as compared to sizes 1 and 3 as we go deeper into the networks. This indicates that it might be beneficial to use larger receptive field sizes in deeper networks.

In summary, the $Q$-learning method enables us to perform analysis on the relative benefits of different design parameters of our state space, and possibly gain insights for new CNN designs.

# E    TOP TOPOLOGIES SELECTED BY ALGORITHM

In Tables A1 through A3, we present the top five model architectures selected with Q-learning for each dataset, along with their prediction error reported on the test set, and their total number of parameters. To download the Caffe solver and prototext files, please visit *https://bowenbaker.github.io/metaqnn/*.

| Model Architecture | Test Error (%) | # Params ($10^6$) |
|---|---|---|
| [C(512,5,1), C(256,3,1), C(256,5,1), C(256,3,1), P(5,3), C(512,3,1), C(512,5,1), P(2,2), SM(10)] | 6.92 | 11.18 |
| [C(128,1,1), C(512,3,1), C(64,1,1), C(128,3,1), P(2,2), C(256,3,1), P(2,2), C(512,3,1), P(3,2), SM(10)] | 8.78 | 2.17 |
| [C(128,3,1), C(128,1,1), C(512,5,1), P(2,2), C(128,3,1), P(2,2), C(64,3,1), C(64,5,1), SM(10)] | 8.88 | 2.42 |
| [C(256,3,1), C(256,3,1), P(5,3), C(256,1,1), C(128,3,1), P(2,2), C(128,3,1), SM(10)] | 9.24 | 1.10 |
| [C(128,5,1), C(512,3,1), P(2,2), C(128,1,1), C(128,5,1), P(3,2), C(512,3,1), SM(10)] | 11.63 | 1.66 |

Table A1: Top 5 model architectures: CIFAR-10.

| Model Architecture | Test Error (%) | # Params ($10^6$) |
|---|---|---|
| [C(128,3,1), P(2,2), C(64,5,1), C(512,5,1), C(256,3,1), C(512,3,1), P(2,2), C(512,3,1), C(256,5,1), C(256,3,1), C(128,5,1), C(64,3,1), SM(10)] | 2.24 | 9.81 |
| [C(128,1,1), C(256,5,1), C(128,5,1), P(2,2), C(256,5,1), C(256,1,1), C(256,3,1), C(256,3,1), C(256,5,1), C(512,5,1), C(256,3,1), C(128,3,1), SM(10)] | 2.28 | 10.38 |
| [C(128,5,1), C(128,3,1), C(64,5,1), P(5,3), C(128,3,1), C(512,5,1), C(256,5,1), C(128,5,1), C(128,5,1), C(128,3,1), SM(10)] | 2.32 | 6.83 |
| [C(128,1,1), C(256,5,1), C(128,5,1), C(256,3,1), C(256,5,1), P(2,2), C(128,1,1), C(512,3,1), C(256,5,1), P(2,2), C(64,5,1), C(64,1,1), SM(10)] | 2.35 | 6.99 |
| [C(128,1,1), C(256,5,1), C(128,5,1), C(256,5,1), C(256,5,1), C(256,1,1), P(3,2), C(128,1,1), C(256,5,1), C(512,5,1), C(256,3,1), C(128,3,1), SM(10)] | 2.36 | 10.05 |

Table A2: Top 5 model architectures: SVHN. Note that we do not report the *best* accuracy on test set from the above models in Tables 3 and 4 from the main text. This is because the model that achieved 2.28% on the test set performed the best on the validation set.

| Model Architecture | Test Error (%) | # Params ($10^6$) |
|---|---|---|
| [C(64,1,1), C(256,3,1), P(2,2), C(512,3,1), C(256,1,1), P(5,3), C(256,3,1), C(512,3,1), FC(512), SM(10)] | 0.35 | 5.59 |
| [C(128,3,1), C(64,1,1), C(64,3,1), C(64,5,1), P(2,2), C(128,3,1), P(3,2), C(512,3,1), FC(512), FC(128), SM(10)] | 0.38 | 7.43 |
| [C(512,1,1), C(128,3,1), C(128,5,1), C(64,1,1), C(256,5,1), C(64,1,1), P(5,3), C(512,1,1), C(512,3,1), C(256,3,1), C(256,5,1), C(256,5,1), SM(10)] | 0.40 | 8.28 |
| [C(64,3,1), C(128,3,1), C(512,1,1), C(256,1,1), C(256,5,1), C(128,3,1), P(5,3), C(512,1,1), C(512,3,1), C(128,5,1), SM(10)] | 0.41 | 6.27 |
| [C(64,3,1), C(128,1,1), P(2,2), C(256,3,1), C(128,5,1), C(64,1,1), C(512,5,1), C(128,5,1), C(64,1,1), C(512,5,1), C(256,5,1), C(64,5,1), SM(10)] | 0.43 | 8.10 |
| [C(64,1,1), C(256,5,1), C(256,5,1), C(512,1,1), C(64,3,1), P(5,3), C(256,5,1), C(256,5,1), C(512,5,1), C(64,1,1), C(128,5,1), C(512,5,1), SM(10)] | 0.44 | 9.67 |
| [C(128,3,1), C(512,3,1), P(2,2), C(256,3,1), C(128,5,1), C(64,1,1), C(64,5,1), C(512,5,1), GAP(10), SM(10)] | 0.44 | 3.52 |
| [C(256,3,1), C(256,5,1), C(512,3,1), C(256,5,1), C(512,1,1), P(5,3), C(256,3,1), C(64,3,1), C(256,5,1), C(512,3,1), C(128,5,1), C(512,5,1), SM(10)] | 0.46 | 12.42 |
| [C(512,5,1), C(128,5,1), C(128,5,1), C(128,3,1), C(256,3,1), C(512,5,1), C(256,3,1), C(128,3,1), SM(10)] | 0.55 | 7.25 |
| [C(64,5,1), C(512,5,1), P(3,2), C(256,5,1), C(256,3,1), C(256,3,1), C(128,1,1), C(256,3,1), C(256,5,1), C(64,1,1), C(256,3,1), C(64,3,1), SM(10)] | 0.56 | 7.55 |

Table A3: Top 10 model architectures: MNIST. We report the top 10 models for MNIST because we included all 10 in our final ensemble. Note that we do not report the *best* accuracy on test set from the above models in Tables 3 and 4 from the main text. This is because the model that achieved 0.44% on the test set performed the best on the validation set.

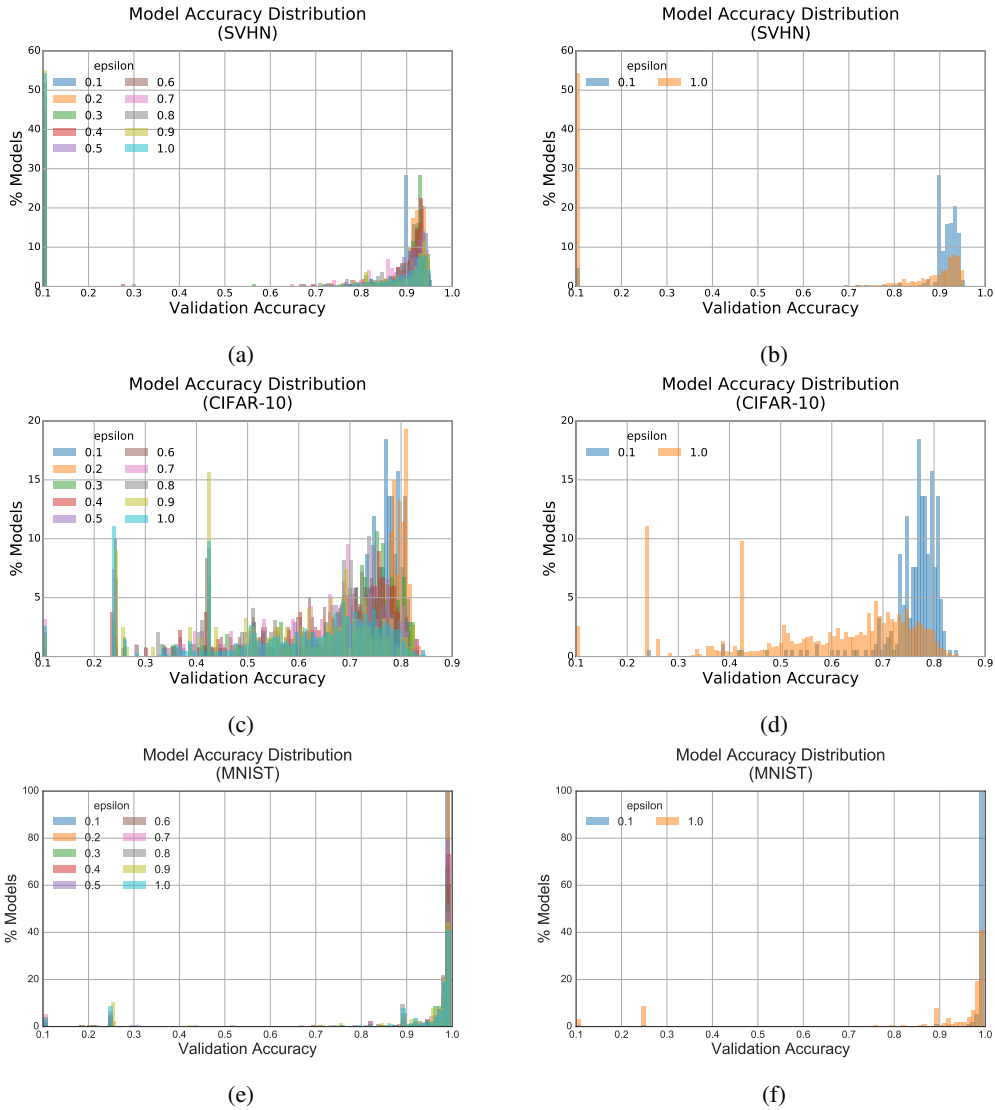

Figure A4: **Accuracy Distribution versus** $\epsilon$: Figures A4a, A4c, and A4e show the accuracy distribution for each $\epsilon$ for the SVHN, CIFAR-10, and MNIST experiments, respectively. Figures A4b, A4d, and A4f show the accuracy distributions for the initial $\epsilon = 1$ and the final $\epsilon = 0.1$. One can see that the accuracy distribution becomes much more peaked in the high accuracy ranges at small $\epsilon$ for each experiment.

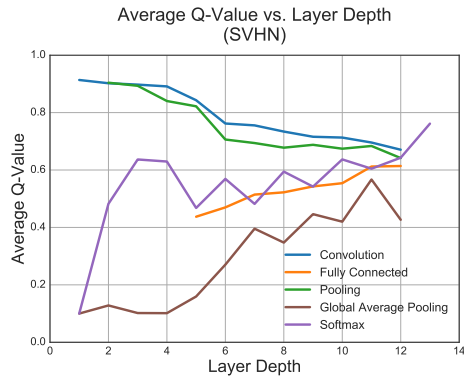

(a)

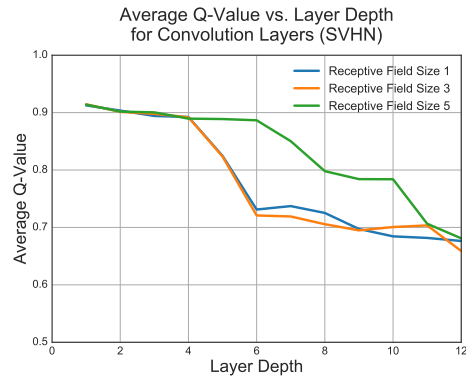

(b)

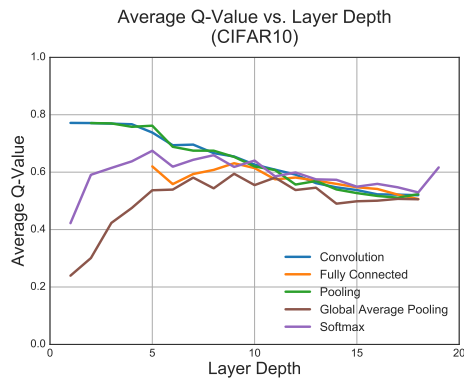

(c)

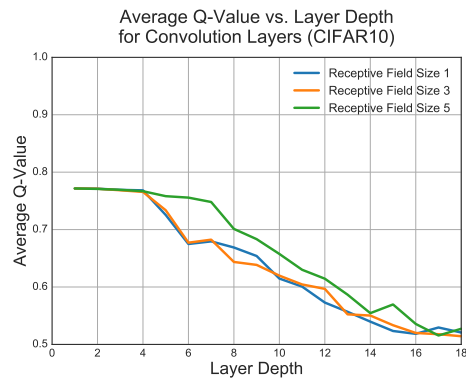

(d)

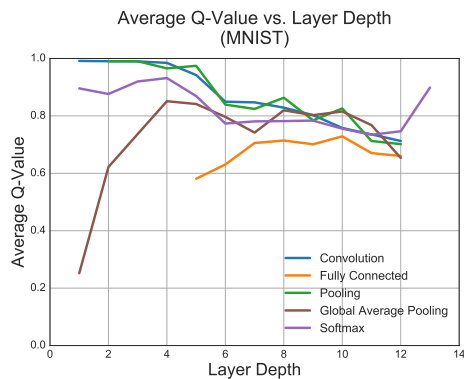

(e)

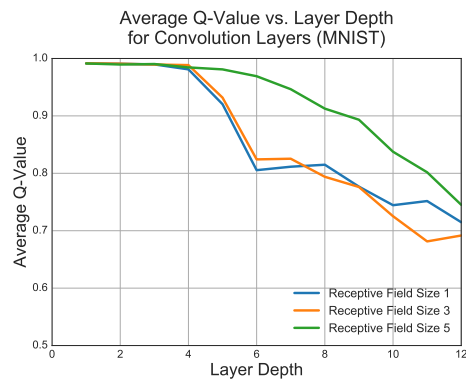

(f)

Figure A5: Average $Q$-Value versus Layer Depth for different layer types are shown in the left column. Average $Q$-Value versus Layer Depth for different receptive field sizes of the convolution layer are shown in the right column.

