# Peer review of "Designing Neural Network Architectures using Reinforcement Learning"

_ICLR 2017 — accepted_

[Official Review · AnonReviewer3 · rating 6 · confidence 4 · 16 Dec 2016]
originality 3 · clarity 4

Authors learn deep architectures on a few small vision problems using Q-learning and obtain solid results, SOTA results when limiting to certain types of layers and competitive against everything else. It would be good to know how well this performs when allowing more complex structures. Paper would be much more convincing on a real-size task such as ImageNet.

[Official Review · AnonReviewer2 · rating 6 · confidence 3 · 19 Dec 2016]
**No Title**
originality 5 · clarity 5 · impact 2 · appropriateness 3

The paper looks solid and the idea is natural. Results seem promising as well.

I am mostly concerned about the computational cost of the method. 8-10 days on 10 GPUs for relatively tiny datasets is quite prohibitive for most applications I would ever encounter.
 I think the main question is how this approach scales to larger images and also when applied to more exotic and possibly tiny datasets. Can you run an experiment on Caltech-101 for instance? I would be very curious to see if your approach is suitable for the low-data regime and areas where we all do not know right away how a suitable architecture looks like. For Cifar-10/100, MNIST and SVHN, everyone knows very well what a reasonable model initialization looks like.

If you show proof that you can discover a competitive architecture for something like Caltech-101, I would recommend the paper for publication.

Minor: 
- ResNets should be mentioned in Table

[Official Review · AnonReviewer1 · rating 6 · confidence 4 · 21 Dec 2016]
originality 3 · impact 4 · meaningful comparison 3

This paper introduces a reinforcement learning framework for designing a neural network architecture. For each time-step, the agent picks a new layer type with corresponding layer parameters (e.g., #filters). In order to reduce the size of state-action space, they used a small set of design choices.

Strengths:
- A novel approach for automatic design of neural network architectures.
- Shows quite promising results on several datasets (MNIST, CIFAR-10).

Weakness:
- Limited architecture design choices due to many prior assumptions (e.g., a set of possible number of convolution filters, at most 2 fully-connected layers, maximum depth, hard-coded dropout, etc.)
- The method is demonstrated in tabular Q-learning setting, but it is unclear whether the proposed method would work in a large state-action space.

Overall, this is an interesting and novel approach for neural network architecture design, and it seems to be worth publication despite some weaknesses.

[Author Response · Nikhil Naik · 20 Jan 2017]
**A new revision updated**

We have added the results of the stability experiment (as suggested by AnonReviewer1) into the Appendix section D.1. We have also included a citation to the CNF paper and results from the latest ResNet paper.

[Final Decision · Program Chairs · 06 Feb 2017]
**ICLR committee final decision**

This paper comes up with a novel approach to searching the space of architectures for deep neural networks using reinforcement learning. The idea is straightforward and sensible: use a reinforcement learning strategy to iteratively grow a deep net graph (the space of actions is e.g. adding different layer types) via Q-learning. The reviewers agree that the idea is interesting, novel and promising but are underwhelmed with the execution of the experiments and the empirical results. 
 
 The idea behind the paper and the formulation of the problem are quite similar to a concurrent submission (